# The Potential Impact of Probiotics on Human Health: An Update on Their Health-Promoting Properties

**DOI:** 10.3390/microorganisms12020234

**Published:** 2024-01-23

**Authors:** Nicoleta-Maricica Maftei, Cosmin Raducu Raileanu, Alexia Anastasia Balta, Lenuta Ambrose, Monica Boev, Denisa Batîr Marin, Elena Lacramioara Lisa

**Affiliations:** 1Department of Pharmaceutical Sciences, Faculty of Medicine, and Pharmacy, “Dunărea de Jos” University, 800010 Galati, Romania; naron@ugal.ro (N.-M.M.); elena.lisa@ugal.ro (E.L.L.); 2Clinic Laboratory Department, Clinical Hospital of Children Hospital “Sf. Ioan”, 800487 Galati, Romania; 3Research Centre in the Medical-Pharmaceutical Field, “Dunarea de Jos” University of Galati, 800010 Galati, Romania; 4Department of Morphological and Functional Sciences, Faculty of Medicine, and Pharmacy, “Dunărea de Jos” University, 800010 Galati, Romania; cosmin.raileanu@ugal.ro (C.R.R.); ambrose.lenu@gmail.com (L.A.); 5Medical Department Faculty of Medicine and Pharmacy, “Dunărea de Jos” University, 800010 Galati, Romania; alexiaanastasia1998@yahoo.com

**Keywords:** probiotics, health, fermented food, microbiome, microbiota

## Abstract

Probiotics, known to be live microorganisms, have been shown to improve or restore the gut microbiota, which in turn has been linked to improved health. It is believed that probiotics are the modern equivalent of a panacea, with claims that they may treat or prevent different diseases both in children and adults (e.g., from colic in babies to cardiovascular disease, respiratory infection, and cancer in adults). Ever since the early 2000s, probiotic-based fermented foods have had a resurgence in popularity, mostly due to claims made regarding their health benefits. Fermented foods have been associated with the prevention of irritable bowel syndrome, lactose intolerance, gastroenteritis, and obesity, but also other conditions such as chronic diarrhea, allergies, dermatitis, and bacterial and viral infections, all of which are closely related to an unhealthy lifestyle. Recent and ongoing developments in microbiome/microbiota science have given us new research directions for probiotics. The new types, mechanisms, and applications studied so far, and those currently under study, have a great potential to change scientific understanding of probiotics’ nutritional applications and human health care. The expansion of fields related to the study of the microbiome and the involvement of probiotics in its improvement foreshadow an era of significant changes. An expanding range of candidate probiotic species is emerging that can address newly elucidated data-driven microbial niches and host targets. In the probiotic field, new variants of microbiome-modulating interventions are being developed, including prebiotics, symbiotics, postbiotics, microbial consortia, live biotherapeutic products, and genetically modified organisms, with renewed interest in polyphenols, fibers, and fermented foods to ensure human health. This manuscript aims to analyze recent, emerging, and anticipated trends in probiotics (sources, doses, mechanism of action, diseases for which probiotics are administered, side effects, and risks) and create a vision for the development of related areas of influence in the field.

## 1. Introduction

One of the glaring issues of the 21st century is the necessity of feeding an ever-growing human population with ever-dwindling natural resources. According to the literature [1], the role of a balanced diet (with the aim of the maintenance of human health) is a principal interest of the scientific community, and numerous studies have demonstrated a reduction in the risk of the appearance of some diseases due to consuming some probiotic-based foods. Due to this, there has been a registered spike in studies that aimed to produce new natural products and in the creation of new products, truly permitting new innovations in the food domain, as well as the creation of new market niches that are tied to functional products. Gut maturation, energy metabolism, nutrition uptake, and the immune system are processes influenced by the gut microbiota and also by our microbiome (billions of microorganisms that form a complex ecology in our gastrointestinal tract) [2]. Recent and ongoing research in microbiome science is opening new research frontiers for probiotics, and clinical research on probiotics will lead to new discoveries related to their influence on human health. These new lines of inquiry will also allow researchers to make new discoveries related to probiotic-based products intended for consumers, the mechanisms of action of these products on health, their nutritional applications, and the obtainment of new probiotics. According to the statements of [3], a new era of significant changes will appear in human health due to the expansion of related fields of interventions aimed at the microbiome and microbiota. In recent years, personalized nutrition (nutrigenetics and nutrigenomics) and precision medicine have also begun to influence the application of probiotics, with a growing interest in modulating the microbial signatures of health and disease. Nutrigenetics deals with the identification of individual genetic variations, which can contribute to the body’s unfavorable responses when we consume certain types of food, and nutrigenomics studies how and what we eat, more precisely, how the bioactive nutrients provided to the body through food can influence our gene behavior, i.e., increase, decrease, or block the activity of genes.

Microorganisms live in and on us in a symbiotic relationship, and it is estimated that close to 95% of these microorganisms are found in our gut (especially the large intestine), whilst the stomach and small intestine are more sparsely colonized. With technological advances in microbiology and gene sequencing, the medical and pharmaceutical industry, food industry, and scientific community have started to better understand the functions of these microorganisms and the metabolites they produce. Since these discoveries, microbiota and microbiome terms have started to become widely used in the groups mentioned above [4].

According to [4], a microbiota is defined as a community of microorganisms, including bacteria, fungi, viruses, and yeasts, present in a defined environment and living on or inside human tissues (skin, lungs, oral mucosa, urogenital tract, gastrointestinal tract). A microbiome can be interpreted as the microbiota (and its genes) living in a given environment; it is regarded as a “living ecosystem”. The gut microbiota is an important participant in the bidirectional exchange of information between the gut and the brain, which is described as the microbiota–gut–brain axis [5].

The term “probiotic” finds its roots in ancient Greek, and it means “for life”. The authors of [6] declared that it was probably Ferdinand Vergin who invented the word “probiotic” in his 1954 article (“Anti-und Probiotika”), where he compared the noxious effects of antibiotics and other antibacterial agents on the intestinal microbiota with the beneficial effects of some useful bacteria [7]. After that, the term “probiotics” was used in 1965 by Lilly and Stillwell, who described them as factors secreted by one microorganism that stimulate the growth of another [8]. Parker [9], in 1974, redefined the term as “organisms and substances which contribute to intestinal microbial balance” and, in 1989, Fuller [10] inaugurated the modern era of probiotics by stating that there are live active cultures used in food fermentation, characterized microbial strains, which are used as additives to food products, and that probiotics have also been used in the form of dietary supplements, ointments, and beauty products (gels, soaps, etc.). Specifically, he referred to probiotics as a “live microbial feed supplement” beneficial to the host animal [9]. On the other hand, in 1998, Ref. [11] stated that it is necessary that we use the appropriate dose of probiotic strains required to achieve a beneficial effect. At present, the most widely accepted scientific definition of probiotics, formulated in 2002 by the FAO (Food and Agriculture Organization of the United Nations) and the WHO (World Health Organization) is “live microorganisms which, when administered in adequate amounts, confer a health benefit on the host” [12]. This definition came from a joint FAO/WHO Expert Consultation, which was requested by the Argentinian Government during an international trade dispute related to powdered milk containing lactic acid bacteria [13]. The FAO/WHO Expert Consultation concluded that the properties and potential benefits of probiotics are dependent on the used dosage and cannot be extrapolated from one microbial strain to another, even if both are part of the same species [14]. In 2013, the International Scientific Association for Probiotics and Prebiotics (ISAPP) supported the same definition [15]. 

Different types of probiotics have different functions, and human health benefits have mainly been demonstrated for specific probiotic strains. Several genera are used as probiotics, including *Lactobacillus*, *Bifidobacterium*, *Bacillus*, *Pediococcus*, and several yeasts [16]. Scientific data show the benefits of specific probiotics in certain types of gastrointestinal diseases: irritable bowel syndrome, the elimination of *Helicobacter pylori*, inflammatory bowel disease, diarrhea, gastrointestinal disorders, allergic diseases (e.g., atopic dermatitis), non-alcoholic fatty liver disease, obesity, insulin resistance syndrome, type 2 diabetes, different types of cancer and the side effects associated with cancer, immune health, metabolic health, dental health, and brain health. Probiotics are also being studied for a potential role as adjuncts in the treatment and prevention of COVID-19.

The aim of this review was to discuss the action mechanisms of probiotics, as well as the current insight into their effect on human health. The selection of probiotic strains, and their respective dosages is crucial in obtaining a therapeutic effect. As such separate, sections of this review are dedicated to this topic. Further research into the acquisition of new probiotic strains, the selection of probiotics dose setting, safety of use, and clinical trials documenting the desired effects on human health is necessary.

## 2. Short History and Uses of Microorganisms as Probiotics

According to academic and popular sources, the history of probiotics began in early civilization, when humans started to consume fermented foods [16]. The first documented uses were around 2000 BC, according to [17]. Although people were not aware of the existence of probiotics, the first food producers turned milk into fermented dairy products using bacteria and yeasts [17]. But [18] declared that our ancestors used yeasts to make beverages far earlier than 2000 BC. As per the statement of [19], as early as 3500 BC, based on artifacts from ancient Egypt, it was found that they were using dairy products such as “Laban Rayad” and “Laban Khed”, (fermented products) which are still popular in the Middle East today. In the ancient Indian Ayurveda literature, it is considered that the use of dairy products and milk is linked to a healthy and long life [20]. Beginning in the early 1900s, Elie Metchnikoff (a Russian research scientist who worked at the Paris Pasteur Institute) suggested that human health could be enhanced by manipulating the gut microbiome (GM) with good bacteria found in yogurt, and this was regarded as the concept of probiotics in medicine [21]. El-Saadony et al. [22] reported that a wide variety of bacteria were identified as the main probiotic candidates in the fish gall bladder, but *Lactobacillus* and *Bacillus* species were of interest due to their strong antagonistic activity and availability. This interest was also related to the biosynthesis of extracellular enzymes commonly used by Ayurvedic doctors in the subcontinent [22].

According to the literature, probiotics are classified into three major groups: *Lactobacilli*, *Bifidobacteria*, and others. 

The *Lactobacillus* genus, described for the first time in 1901, contains species of rod-shaped bacteria that have long been the component of fermented foods and are found in humans as members of the digestive and vaginal cavities. Many currently used probiotics come from the *Lactobacillus* genus. Nicknamed “*lactobacilli*”, this group of microbes is multifunctional, with benefits ranging from extending the shelf life of foods (e.g., making yogurt or cheese from milk) to improving health when administered in the form of probiotic foods and supplements [23]. Scientists have reclassified the genus *Lactobacillus* into 25 genera, including the emended genus *Lactobacillus* (*L. delbrueckii* group and *Paralactobacillus*) and 23 novel genera: *Acetilactobacillus*, *Agrilactobacillus*, *Amylolactobacillus*, *Apilactobacillus*, *Bombilactobacillus*, *Companilactobacillus*, *Dellaglioa*, *Fructilactobacillus*, *Furfurilactobacillus*, *Holzapfelia*, *Lacticaseibacillus*, *Lactiplantibacillus*, *Lapidilactobacillus*, *Latilactobacillus*, *Lentilactobacillus*, *Levilactobacillus*, *Ligilactobacillus*, *Limosilactobacillus*, *Liquorilactobacillus*, *Loigolactobacilus*, *Paucilactobacillus*, *Schleiferilactobacillus*, and *Secundilactobacillus* [23]. This new *Lactobacillus* taxonomic classification means the species that are more closely related or based on shared physiological and metabolic properties belong to the same genus. In turn, this may facilitate our understanding of common mechanisms that could mediate probiotic health benefits [23].

*Bifidobacterium* spp. is a very important group of probiotics for humans because it can reproduce and metabolize in the middle and end of the small intestine and large intestine and secrete bifidogenic factors with probiotic effects to regulate intestinal health and adapt to the anaerobic intestinal life [24,25]. The physiological functions of *Bifidobacterium* are mainly as follows: restraining the growth of pathogenic bacteria, inhibiting proinflammatory cytokines [26,27], protecting against intestinal barrier dysfunction (in vitro and in vivo), synthesizing vitamins and amino acids in the intestine and increasing calcium bioavailability [28,29], and antitumor effects, according to experimental results suggesting that *Bifidobacteria* can be special carriers of anti-cancer proteins against malignant tumors [30].

Other bacteria species such as *Enterococcus* are frequently used in the food industry today and present a key feature, which is the ability of the strains to compete, survive, and attach to host cells in the intestine [31]. Also, they are highly resistant to a wide range of pH and temperature values (these characteristics are attributed to their strong bacteriocin production capacity) and can be used as a natural antimicrobial agent in the food industry [31].

*Saccharomyces cerevisiae* is a well-known non-pathogenic and selective probiotic strain that is now used in the commercial production of probiotic foods. *Saccharomyces* strains have been extensively studied for their probiotic action and are commonly used to treat digestive disorders (diarrhea symptoms), especially when used as an adjunct to antibiotic therapy. This is because, when passing through the digestive tract, *Saccharomyces* strains have a higher survival capacity compared with the other probiotics. Also, these strains help to maintain the balance of the normal microflora of the intestinal tract and have immunomodulatory effects, acting to fine-tune immunological pathways during pathogenic infections or chronic diseases [32,33,34].

Other categories of probiotics that are more commonly used are *Bacillus* spp., *E. coli*, and *Streptococcus* spp. Based on recently published work, their roles and applications are the same as some common probiotics [35] (see Table 1).

The number of probiotics is measured in colony-forming units (CFUs), which indicates the number of viable cells—for example, 1 × 10^9^ or 1 billion CFU and 1 × 10^10^ or 10 billion CFU). Many probiotic supplements contain 1 to 10 billion CFU per dose, but they are very important strain designations in the clinical setting because they make the connection between clinical benefits (e.g., the prevention of certain specific types of diarrheas) and both specific strains and mixtures of specific strains in effective doses. Often, some strains have unique properties [80] that may be linked to their specific medical benefits [81], but evidence is emerging that some mechanisms of probiotic activity are shared among different strains, species, and genera. Worldwide, probiotic products are available in the form of dietary supplements (including tablets), capsules, powders, liquids, and other formulations (cosmetics for skin: cream, deodorant, gel, mask, foundation, serum, soap bar). They are also added to commercial products including food, beverages, bakery products, sweets, etc.

## 3. Colonizing Microbiota and Factors Affecting Microbiota

The formation of the intestinal microbiota undergoes a staged evolution. Bacterial colonization has recently been considered to begin in utero because of the presence of bacteria in the placental tissue, fetal membranes, amniotic fluid, and umbilical cord blood in healthy women and newborns, which suggests early contact with the bacterial world. Colonization of the digestive tract is continued and amplified after birth when the intestine of the newborn begins to be colonized by microbes originating from:Contact with the mother’s chair;Vagina, skin, and breast milk;The environment.

Vaginal birth induces an intestinal population that reflects the structure of the maternal vaginal flora, whereas the intestinal microbiota of cesarean-born babies is dominated by the maternal skin and oral flora, but also includes the skin ecosystem of the medical staff and germs coming from other surfaces. Also, breast milk contains live bacteria from the mammary ducts and areola, but especially from the maternal intestine, which is a source of natural colonization of the neonatal intestine. The physiological bacterial population of the intestine modulates the expression of epithelial genes for immunological tolerance, provides nutrients to the host, and contributes to the maturation of systemic immunity. Initially, these microbes develop in a relatively simple ecosystem dominated by the *Bifidobacteria*. The complexity of this ecosystem increases and evolves in the first 2 years of life when the microbiota is fully developed, and the predominant microorganisms are part of two phyla: *Bacteroidetes* and *Firmicutes*. However, there are many factors that can affect the initial establishment of the microbiota, its composition, and its diversity later in life [4]. In early life, factors that affect the microbiota are preterm delivery/full term; vaginal/cesarean section delivery; breastfeeding/formula feeding; and having/not having pets. Later in life, such factors are diet; consumption of fermented foods, probiotics, and prebiotics; physical activity and time spent indoors or outdoors; living in rural or urban areas; and the use of drugs (antibiotics and proton pump inhibitors) [4].

The mechanism of action of probiotics relates to their influence on the microbes that inhabit the gastrointestinal (GI) tract. Wang et al. [82] declared that approximately 100 trillion microorganisms, such as bacteria, viruses, fungi, and protozoa, of at least 1000 different species live in the GI tract [82]. Very few are found in the stomach and duodenum, where gastric acid, bile, and pancreatic juices either inhibit or eradicate most microorganisms. Lower down in the intestine, the numbers of microbes progressively increase, from 10^4^ cells per gram in the jejunum to 10^9^ cells per gram in the distal ileum. The most heavily populated region of the GI tract is the colon, mainly with anaerobic microbes, and it contains up to 10^12^ cells per gram of intestinal content [83]. Normal microbial colonization of the human body is dependent on the conditions of the local chemical environment, the degree of oxygenation, the nutritional intake of the host tissue, and the intervention of the immune system, and for these reasons, the mechanisms of action of probiotics are complex and often strain-specific. The benefits of a probiotic are the result of the interaction between the probiotic strain, the resident microbiota, and the host. These interactions can be metabolic, microbiological, physiological, neurological, endocrinological, and immunological in nature, including different combinations thereof. Studies have shown that certain probiotic microbes can help promote resistance and normalize the gut microbiota when it has been disrupted by antibiotics or other stressors, even though it is quite difficult for scientists to define exactly what a healthy microbiome is [81].

Wang et al. [82] reported that populations of colonizing microbes differ between healthy individuals and others with disease or poor health, and for this reason, the microbial composition appears to differ according to age, sex, race, and geographical location. Taking these things into account, the diversity of gastrointestinal microbes between individuals is very different. Each person harbors his or her own distinctive pattern of microbial composition, and this is determined by genotype, initial colonization at birth, and dietary habits [82].

Probiotics interact with intestinal bacteria after entering the intestine. The intestinal mucosa plays the first role of a physical barrier, which keeps the intestine at a safe distance from toxic substances in the intestinal lumen. After that, probiotics react with bacteria following their entrance into the intestine, enhancing the chemical barrier, biological barrier, immune barrier, and mechanical barrier [84]. Once arrived in the intestine, they interact with intestinal cells, with the aim of restoring intestinal permeability, stimulating mucus production, promoting mucosal regeneration, maintaining the intestinal mechanical barrier’s normal function, and maintaining the mucosal barrier’s integrity [85].

## 4. Mechanisms of Action of Probiotics

In recent years, significant progress has been observed in the field of probiotics studies, especially in terms of selection and the characteristics of individual probiotic cultures and their effect on health.

There are several mechanisms known to explain the mode of action of probiotics against diseases because probiotics can transiently colonize the human gut in individualized patterns, depending on the individual’s microbiota, the probiotic strain/strains, and the region of the GI tract [86].

According to data in the literature, the advancement of the epithelial barrier, increased adhesion to the intestinal mucosa, production of antibacterial substances, competitive exclusion of pathogens, simultaneous suppression of bacterial adhesion, and modulation of the immune system are some of the main mechanisms of action of probiotics [87,88,89,90,91] (Figure 1).

Probiotics interact with the gastrointestinal ecosystem by inhibiting the growth of pathogenic microorganisms (e.g., accelerating gastrointestinal transit and reducing the ability of disease-causing bacteria to colonize and adhere to the gastrointestinal mucosa). Probiotics also increase the production of bioactive metabolites (e.g., short-chain fatty acids—SCFAs), thus lowering the pH in the colon. Other modes of action include vitamin synthesis in the gastrointestinal tract, bile salt metabolism, enzyme activity and toxin neutralization [80], the up-regulation of intestinal electrolyte absorption, the regulation of lipid metabolism, and the suppression of the growth of pathogenic bacteria by directly binding to Gram-negative bacteria. Probiotics communicate with cells inside and outside the GI tract through biochemical signaling mechanisms, and this leads to improved intestinal barrier function, reduced production of pro-inflammatory markers such as cytokines [92], and improved immune function [93]. 

The mucus layer and also the intestinal epithelium have their own specific cell types, which can be a physical barrier to intestinal microorganisms. For example, Paneth cells can synthesize and secrete antimicrobial peptides during contact with intestinal bacteria, *Saccharomyces cerevisiae* cells can secrete mucus, enterocytes can absorb molecules from the intestinal lumen, and intestinal endocrine cells are a constituent part of the intestinal epithelium [94,95,96]. The main roles of the intestinal epithelial mucus layer include developing a protective barrier against the hostile luminal environment, facilitating the passage of food, and avoiding the adhesion of pathogenic agents to their own lamina. The first element that plays the role of a physical barrier is the intestinal mucosa, which has the role of keeping the intestine at a safe distance from toxic substances in the intestinal lumen. After entering the intestine, probiotics interact with bacteria to enhance their chemical, mechanical, biological, and immune barriers [84], and once in the gut, probiotics interact with intestinal cells. The main goals are to restore intestinal permeability, promote mucosal regeneration, stimulate mucus production, and maintain mucosal barrier integrity and normal intestinal mechanical barrier function [85]. Through these mechanisms, probiotics can have a much greater influence on diseases and human health (children and adults).

## 5. Probiotics and Human Health

The effect of probiotics on human health has been studied in a wide range of conditions, and many studies focused on gastrointestinal conditions, atopic eczema, allergies, respiratory tract infections, obesity, metabolic disease/type 2 diabetes, cardiovascular, cognitive, and mental diseases, bone health, nonalcoholic fatty liver disease (NAFLD) and hepatic encephalopathy, tumor necrosis factor (TNF-α), autism, burn wounds, and gynecological diseases. Regarding the administration of probiotics, the guidelines specify the following rules: ➢Probiotics must be administered on an empty stomach.➢Heat-dried formulations must be kept in the refrigerator (4 °C), and lyophilized forms can be kept at room temperature.➢A probiotic is administered one to two times a day because the dosage varies from 1 to 10 billion (10^8^–10^9^) CFUs.➢For patients who take antibiotics, the dosing time must be separated with an interval of two hours between them [97].

### 5.1. Probiotics and Treatment of Infectious Burn Wounds

The first barrier against microorganisms is the skin, but if it is removed, the lesions are continuously exposed to pathogens and Gram-positive bacteria (*Staphylococcus aureus*) of the normal skin flora, which are the first cause of infection. Also, in recent years, the Gram-negative bacteria *Pseudomonas aeruginosa* has been counted as one of the most prevalent causes of wound infection. Physicians and pharmacists sought alternative therapies for controlling wounds, and novel solutions—probiotics—were discovered; these new substances promoted the growth of beneficial microorganisms. The benefits of probiotics in preventing and treating skin infections were demonstrated in clinical studies. As an effective treatment for wounds, they are recognized as topical probiotics because they secrete antimicrobial substances (i.e., hydrogen peroxide [98] and bacteriocin [99]), which inhibit the growth of pathogenic microorganisms and prevent wound infection.

Farahani et al. [100] studied probiotic-containing dressings to tackle burn injury concerns like infection and microbial resistance. Their study analyzed the effects of probiotic-loaded microparticles with in situ gelling properties on infectious burn wounds. A probiotic strain with proven wound-healing properties was chosen: the *Lactiplantibacillus plantarum* strain. The researchers declared that the experiments revealed the probiotic-loaded microparticles were more effective and improved the management of burn wound infections. Also, histological investigations showed that the probiotic-loaded particles functioned exactly as efficaciously as in the case of silver sulfadiazine ointment. However, despite the effectiveness of probiotics in burn wound healing, their administration and stabilization are challenging because protection against Gram-positive and Gram-negative bacteria requires the use of different forms of dosage such as hydrogels [101], films [102], ointments [103], and gels [104]. A proper way to stabilize a probiotic strain could be by encapsulating them in polymeric particles and natural polymers, for example, sodium alginate and chitosan have been widely investigated.

### 5.2. Probiotics and Gastrointestinal Diseases

There is substantial evidence in the literature of the beneficial effects of probiotics on diarrhea, especially the prevention of traveler’s diarrhea [105,106] and the treatment of acute diarrhea in children [107]. *Saccharomyces boulardii* and *Lactobacillus rhamnosus* GG are the most frequently used strains of probiotics and are associated with successful results in diarrheal diseases. However, following a Cochrane analysis of 82 studies that included adults and children, the conclusion was that probiotics still have a low, even uncertain, ability to reduce the duration of acute diarrhea [108].

Also, *S. boulardii* and *L. rhamnosus* GG have been shown to reduce the risk of Antibiotic-Associated Diarrhea (AAD). Both probiotic strains have been shown to be effective in preventing AAD [109,110,111] and/or *Clostridium difficile* diarrhea [112,113]. The researchers in these studies reported that higher doses are the most effective in these conditions. 

It is mentioned in the literature that strains of *S. boulardii* and *L. rhamnosus* GG have been used to treat inflammatory bowel disease (IBD) along with various other strains of *lactobacilli*, *bifidobacteria,* and *Streptococcus thermophilus*. However, to date, the efficacy of probiotic strains in maintaining remission in IBD is still unclear [114,115], and although six recent meta-analyses [114,115,116,117,118] showed that probiotic strains induce remission in IBD, with much evidence of efficacy for ulcerative colitis (UC) but less evidence for Crohn’s disease (CD) [116,117]. There is also evidence for the usage of certain strains of probiotics in preventing an attack of pouchitis and also in preventing subsequent attacks after remission, which have been induced.

In terms of managing the symptoms of irritable bowel syndrome (IBS), including predominant constipation, probiotics have little benefit. Most reported studies used *Lactobacilli* and *Bifidobacteria*. However, in a meta-analysis of 15 randomized controlled trials (RCTs), consumption of probiotics substantially reduced gastrointestinal transit, improved stool consistency, and increased stool frequency [119].

Regarding lactose intolerance, variable but generally favorable effects were demonstrated when probiotics were administered [120,121].

In accordance with the literature, there is evidence that probiotic strains can be effective as adjuncts to antibiotic therapy in eradicating *Helicobacter pylori* in patients with peptic ulcers [122]. Wilkins and Sequoia [123] stated that in a study involving 1163 children and adults, they found that the use of *Lactobacillus*-based probiotics as an adjunct to antibiotics increased the eradication rate of *H. pylori* compared with a control.

Although the quality of the studies is generally poor [124,125], evidence has been reported showing that probiotic strains can help prevent necrotizing enterocolitis in preterm infants.

There is also evidence that probiotics can alleviate NAFLD [126], and they appear to be effective in the treatment of hepatic encephalopathy. A meta-analysis of six RCTs involving 496 adults with cirrhosis showed that probiotic therapy significantly reduced the development of overt hepatic encephalopathy [127].

### 5.3. Probiotics in the Treatment of Dermatological Diseases

Reviewing the literature, we noticed that of the studies carried out in the last five years, only four studies evaluated treatment with probiotics during pregnancy, breastfeeding, and/or childhood, and in all these studies, the topic was the effectiveness of probiotic strains in reducing the risk of atopic eczema in children [128,129,130,131]. Following these studies, it was mentioned that only some strains of *Lactobacillus* spp., *Bifidobacterium* spp., and *Propionibacterium* spp. showed beneficial results, while other strains of the same species did not.

The administration of probiotics has also been used in the treatment of acne as adjunctive therapy. In vitro investigations have shown that certain strains such as *L. salivarius* LS03, *Lactococcus*, and *Streptococcus salivari* produce bacteriocins that inhibit the growth of *Cutibacterium acnes* [132]. In addition, *B. adolescentis* SPM0308, due to antimicrobial activity, proved to be effective in controlling the growth of both *C. acnes* and *S. aureus* [132].

Also, *S. thermophiles* has been shown to be beneficial in the production of lipids (ceramides in the stratum corneum), which help the skin to maintain moisture, and also phytosphingosine, which helps fight *C. acnes* [132].

For the treatment of acne vulgaris, [133] used the strain of *Weissella viridescens* UCO-SMC3, and with topical and oral administration, obtained a favorable modulation of the inflammatory response. After their study, the researchers found that the volunteers who participated in the study had reduced numbers of acne lesions after applying the cream. It should be mentioned that [133] were the first researchers to characterize the antimicrobial and immunomodulatory characteristics of the *W. viridescens* UCO-SMC3 strain, which is a bacterium that was isolated from garden slime (*Helix aspersa* Müller).

Also, there are clinical studies in which the effects of oral probiotics on psoriasis were described, and these effects were found to significantly improve the condition of the patients [134]. In an RCT, Ref. [135] stated that probiotics significantly reduced the doctor’s global assessment index. Ninety patients participated in the study and were randomized into two groups (probiotics and placebo) with a duration of 12 weeks, and a reduced risk of relapse was observed six months after the end of the study.

### 5.4. Probiotics and the Study of Obesity and Cardiometabolic Parameters

Research indicates that disruptions in the composition of the gut microbiota can disrupt its function and improve the gut barrier, resulting in low-grade chronic inflammation. With the appearance of inflammation, the effects are considerable for the adiposity of the host and also insulin resistance. For this reason, the fundamental role of gut microbiomes in appetite regulation and the resulting effect on obesity is now clear [136], with studies indicating that probiotics can contribute to weight loss and an improvement in cardiometabolic parameters. The literature presents studies with positive results regarding obesity, and the probiotic strains most often used are *Lactobacillus* strains. 

Systematic reviews have shown that by consuming probiotics [137,138,139,140], overweight and obese people lose weight, but the weight loss and reduction in waist circumference and visceral fat is small [137,138,139,140]. In another meta-analysis study, it was shown that the probiotic strains *B. breve*, *B. longum*, *S. thermophilus*, *L. acidophilus*, *L. delbrueckii*, and *L. casei* can have a positive impact on anthropometric results and also on the metabolic risk factors (e.g., fasting glucose in individual glucose or insulin) in subjects with metabolic diseases [141]. Researchers have stated that specific mechanisms mediated in the gut by ingested probiotics may be involved (for example, digestion and absorption of sugar or increased production of short-chain fatty acids) [142]. 

Studies also show that the consumption of probiotics can improve inflammatory markers (C-reactive protein) and glycemic control and insulin metabolism in patients diagnosed with type 2 diabetes or metabolic syndrome [143,144,145,146,147], including pregnant women with gestational diabetes [143,144]. On the other hand, it has been shown that the administration of probiotic markers of cardiovascular diseases can be improved, including hypertension [148] and dyslipidemia [149,150]. Because the studies presented in these systematic reviews tend to lack homogeneity, it is difficult at this time to clearly state the benefits of probiotics in obesity and cardiometabolic parameters. The results clearly highlight the need for further studies, preferably conducted as multicenter, randomized, placebo-controlled trials, to determine whether the two pathologies (obesity and insulin resistance) can be ameliorated by increasing the colonization or activity of the administered probiotic or other gut microbes that are associated with thinness and the lack obesity.

### 5.5. Probiotics in Viral Infections

This section will explore the evidence for probiotics in several key viral infections including SARS-CoV-2 infection, viral upper respiratory tract infections (URTIs), influenza infection, viral hepatitis, human immunodeficiency virus (HIV), human papilloma virus (HPV), and vaccination.

#### 5.5.1. Severe Acute Respiratory Syndrome Coronavirus 2 (SARS-CoV-2) Infection and Probiotics

Previous studies have shown a beneficial effect regarding the administration of probiotics for the treatment of respiratory infections, probably due to their immunomodulatory and anti-inflammatory effects. In this context, probiotic strains were also investigated during the SARS-CoV pandemic. To date, clinical trials investigating the potential therapeutic role of probiotics in relation to SARS-CoV-2 have been published, and in most studies, the probiotics used consist of lactic bacteria including *Lactobacillus* and *Bifidobacterium* strain, as these bacteria have been promoted as potential immunomodulators. In the studies published to date, the main findings included improvements in symptoms following the ingestion of probiotics, indicating that probiotics appear to be more efficient in reducing levels of fatigue and possibly the resolution of gastrointestinal problems. A study by [151] reported improvements in patients who received probiotics and enzyme supplements, as 91/100 patients with SARS-CoV-2 who received the treatment with probiotics (*Bacillus coagulans*, *Bacillus subtilis* and *Bacillus clausii*) showed diminished physical and mental fatigue compared with 15/100 placebo patients on day 14 of the treatment. After the administration of a probiotic based on *L. plantarum* KABP022, KABP023, and KAPB033 and *Pediococcus*, [152] observed faster recovery and shorter durations of symptoms in a blinded RTC versus placebo. In another study, [153] reported that in a randomized, placebo-controlled trial, the gastrointestinal symptoms of the tested SARS-CoV-2 patients appeared to improve, and hospital-acquired diarrhea was less frequent in patients who received a probiotic compared with those who received a placebo. Also, [154] reported that in two studies, the opportunistic pathogens *Actinomyces*, *Erysipelaclostridium*, *Streptococcus*, *Veillonella*, *Rothia*, and *Enterobacter* were associated with the diagnosis and/or severity of COVID-19, and the beneficial butyrogenic bacteria *Faecalibacterium* and *Anaerostipes*, as well as *Bifidobacterium*, were inversely correlated with the diagnosis and/or severity of COVID-19.

#### 5.5.2. Viral Upper Respiratory Tract Infections and Probiotics

The upper respiratory tract is composed of the nostrils, nasal cavity, oral cavity, tonsils, pharynx, and larynx, and URTI is the term that encompasses several pathogenic conditions that affect these structures (for example, the common cold—which mainly affects the nose—tonsillitis, pharyngitis, laryngitis, acute otitis media, and sinusitis). The most common viral URTIs are caused by rhinoviruses, respiratory syncytial viruses, adenoviruses, influenza viruses, parainfluenza viruses, coronaviruses, and human metapneumovirus [155]. Hao et al. [156] stated that a significant number of clinical trials have examined the efficacy of probiotics in the prevention of URTIs, and probiotics are believed to prevent respiratory tract infections in similar ways to the prevention of gastrointestinal (GI) infections, namely, by modulating local and systemic immunity. Specific effects may include increased phagocytic activity of peripheral leukocytes, increased secretion of immunoglobulins (IgA, IgG, and IgM), and increased production of cytokines: interleukins, TNF-a, and interferon-α (IFN). In a Cochrane review, the same researchers [156] examined 12 RCTs of probiotics for the prevention of URTIs, involving a total of 3720 participants. The studies included data from all age groups: children, adults, and the elderly, from the following continents: Europe, North America, South America, and Asia. The researchers stated that probiotics were better than the placebo based on measurements of the number of participants experiencing acute URTI episodes, the average duration of an acute URTI episode, the reduced rates of antibiotic prescriptions for acute URTIs, and the number of school absences due to colds [156]. Poon et al. [157] conducted a meta-analysis study in which they analyzed the strain *L. paracasei* subsp. *paracasei* CNCM I-1518 and concluded that there appears to be a promising therapeutic benefit to taking probiotics for the prevention of URTIs.

#### 5.5.3. Influenza Infection and Probiotics

*Human influenza* A, B, and C viruses primarily attach to and replicate within respiratory epithelial cells that line the upper to lower respiratory tract [158]. It has been found that the flu usually causes only mild and uncomplicated illness, and most patients recover without the need for medical intervention. Risk factors associated with disease severity include extreme age and patients with comorbidities (chronic cardiorespiratory diseases) [158]. Belkacem et al. [159] used the probiotic strain *L. paracasei* CNCM I-1518 in a rodent model of influenza infection and reported that the ingestion of the strain reduced susceptibility to influenza infection, reduced inflammatory cell infiltration into the lungs, and increased the rate of viral clearance. Additional studies in rodents have shown that the microbiome plays an important role in mediating IFN signatures in the lung tissue by inducing an early refractory environment for influenza virus replication, thus reducing early infection with the influenza virus [160]. Wu et al. [161] stated that antibiotic-induced dysbiosis in rodents can reduce the expression of receptor 7 (TLR7) and NF-kB mRNA, leading to impaired antiviral immunity, which is reversed with the administration of probiotic strains of *Bifidobacterium* and *Lactobacillus* (sensu lato).

#### 5.5.4. Viral Hepatitis and Probiotics

Hepatitis B (HBV) and C (HCV) infections are global health problems, particularly in developing countries, and the pathogenic interaction between the virus and the host’s immune system can lead to liver damage and potentially cirrhosis and hepatocellular carcinoma. Lee et al. [162] reported that in the in vitro cell model HepG2.2.15, containing integrated HBV DNA and secreting hepatitis B virus surface antigen-AgHBs, the cell extract of *B. adolescentis* SPM0212 inhibited HBV, and its antiviral mechanism was associated with the Mx GTPase pathway. The Mx GTPase pathway is one of the four main effector pathways in the IFN-mediated antiviral response. Interestingly, [163] reported that a specific species, *S. salivarius*, was dramatically increased in association with chronic hepatitis C (CHC) progression. 

Research has also shown a significant increase in *Alcaligenaceae*, *Porphyromonadaceae*, *Veillonellaceae*, *Enterococcus*, *Megasphaera*, and *Burkholderia* in the GI microbiome of cirrhotic patients with hepatic encephalopathy (HE), and this taxonomic shift has been associated with hyperammonemia and systemic inflammation [164]. In a prospective RCTs, it was shown that probiotics are effective in the primary prevention of HE in patients with cirrhosis [165] because probiotics can act on pro-inflammatory urease-producing pathogenic organisms and thus preventing the development of HE.

#### 5.5.5. HIV and Probiotics

One of the key aspects of HIV infection is the rapid and extensive destruction of CD4+ T lymphocytes [166], which becomes more exacerbated in the later stages of infection [167,168]. Probiotic interventions have become an interesting therapeutic target for HIV patients because they promote tolerogenicity [169], can replace pathogenic strains, and reduce inflammation [170]; consequently, T cell depletion and senescence could be reduced. This opens an avenue for an improvement in immune reconstitution after viral suppression. Blázquez-Bondia et al. [171] conducted a study to test the safety, tolerability, and effectiveness of a probiotic consisting of two strains of *L. plantarum* and one of *Pediococcus acidilactici*, combined with prebiotic fibers, in patients with HIV and reported that it was safe and well-tolerated. 

Also, the status of the vaginal microbiome has an important relevance to HIV. Probiotic bacteria (*Bifidobacteria* and *L. rhamnosus*) can increase the production of short-chain fatty acids and vitamins, modify the intestinal epithelium and immunity, improve intestinal barrier function, help prevent bacterial vaginosis, and reduce disease burden [172]. Another study investigated the therapeutic potential of probiotics in reducing the incidence of diarrhea after administration of a single dose of probiotics (1 × 10^9^ CFU/mL) for 30 days in HIV patients. Researchers found that it takes more than 30 days to see the beneficial effects of the probiotics [173]. Therefore, it is recommended that people afflicted by HIV should take probiotics frequently as dietary supplements, but care should be taken when prescribing probiotic supplementation to HIV patients [174].

#### 5.5.6. HPV and Probiotics

Probiotics have been studied in the context of HPV following research that aimed to improve genital viral clearance and cervical smear quality. A prospective study of 54 women with HPV and a diagnosis of low-grade squamous intraepithelial lesion in PAP smear was followed for 6 months [175]. The researchers concluded that after the daily consumption of a probiotic drink, HPV was cleared in 29% of the patients using the probiotics compared with 19% of the control patients [175]. In another double-blind RCT study involving 121 women with genital high-risk HPV infection (HR-HPV), researchers reported that the application of two strains of probiotics, *L. rhamnosus* GR- 1 and *Limosilactobacillus reuteri* RC-14, for 3 months did not influence HR-HPV genital clearance, but a decrease in the rates of abnormal and unsatisfactory cervical smears was obtained [176]. Very interestingly, it has been shown that the long-term addition of vaginal probiotics has a superior ability to reduce cytological abnormalities and improve HPV clearance compared with the short-term administration of probiotics [177]. However, it seems that the functional support of probiotics and the eventual restoration of the protective vaginal microbiota are critical elements for successfully combating HPV infections regardless of the treatment approach or duration [154].

#### 5.5.7. Vaccine and Probiotics

Vaccination works to prepare the adaptive immune system to recognize pathogens before infection. This allows for a quick response when and if further exposure occurs [154]. It has been reported in the literature that in an animal study, certain strains of probiotics determined the enhancement in the immune response to a vaccine and also reduced the risk of subsequent infections. Regarding human studies, a growing number of well-controlled studies have indicated that the response to vaccines against influenza, cholera, and other childhood diseases was improved with selected probiotics [178]. In support of this, a recent systematic review of human clinical trials regarding the influence of probiotics on vaccine response was demonstrated by [179]. The researchers found evidence suggesting that certain probiotics can enhance the immune system’s response to influenza vaccination, which increases the potential benefit for the elderly, in whom the response to influenza vaccination (seroconversion) is impaired [179]. Yeh et al. [180] conducted a clinical trial using probiotics as adjuvants for influenza vaccination, and the titers of hemagglutination inhibition (HI) antibodies were examined. In this meta-analysis, researchers reported significantly higher HI titers for both A/H1N1, A/H3N2, and B strains, with increases of 20%, 19.5%, and 13.6%, respectively, in the probiotic group compared with the control. These data suggest the promising role of the microbiome in enhancing the host’s antiviral immunity. Also, in a meta-analysis study, researchers reported that, in adults who received a vaccine to protect against influenza viruses, the administration of probiotics was effective [181].

## 6. Guidance When Choosing a Probiotic Product

Choosing the best probiotic to administer to patients will continue to be a shifting target, given the fact that more and more research and clinical trials are being performed. Data from the literature indicates that the first aspect is related to the mode of therapy, prevention, or treatment, because different probiotics may be more suitable for each objective [182]. The second aspect refers to the fact that certain probiotics are more suitable for the pediatric population compared with the adult population. Also, the most suitable single-strain or probiotic mixture can then be selected based on knowledge and the summary of clinical studies reported in the literature [182]. Depending on availability and regulatory oversight in each country, other considerations may also assist in selecting an appropriate probiotic. Probiotics available as dietary supplements should list the manufacturer, the probiotic strains in the product, and the daily dose and comply with health/function claims as per each country’s regulations. Products that do not include these indications should be viewed with caution, and if the probiotic is regulated as a prescription drug, the advice of a knowledgeable pharmacist or physician should be sought out.

Regarding infection management, the guideline for the management of gastrointestinal disorders was formulated by the American Gastroenterological Association (AGA). The following is a summary of the recommendations according to the AGA: -For infection with *Clostridioides difficile*, the use of probiotics only in the context of a clinical trial is recommended.-For the case of adults and children on antibiotic treatment, the use of *S boulardii*; the two-strain combination of *L acidophilus* CL1285 and *L. casei* LBC80R; the three-strain combination of *L. acidophilus*, *Lactobacillus delbrueckii* subsp. *bulgaricus*, and *B. bifidum*; or the four-strain combination of *L. acidophilus*, *L. delbrueckii* subsp. *bulgaricus*, *B bifidum*, and *S. salivarius* subsp. *thermophilus* over no or other probiotics for prevention of *C. difficile* infection is recommended.-For adults and children with Crohn’s disease and ulcerative colitis and symptomatic children and adults with irritable bowel syndrome, the use of probiotics only in the context of a clinical trial is recommended.-For the case of adults and children with pouchitis, the eight-strain combination of *L. paracasei* subsp. *paracasei*, *L. plantarum*, *L. acidophilus*, *L. delbrueckii* subsp. *bulgaricus*, *B. longum* subsp. *longum*, *B. breve*, *B. longum* subsp. *infantis*, and *S. salivarius* subsp. *thermophilus* over no or other probiotics is recommended.-For children with acute infectious gastroenteritis, the use of probiotics is not recommended.-For the case of preterm (less than 37 weeks gestational age), low-birth-weight infants, the use of a combination of *Lactobacillus* spp. and *Bifidobacterium* spp. (*L. rhamnosus* ATCC 53103 and *B. longum* subsp. *infantis*; *L. casei* and *B. breve*; *L. rhamnosus*, *L. acidophilus*, *L. casei*, *B. longum* subsp. *infantis*, *B. bifidum*, and *B. longum* subsp *longum*; *L. acidophilus* and *B. longum* subsp. *infantis*; *L. acidophilus* and *B. bifidum*; *L. rhamnosus* ATCC 53103 and *B. longum* Reuter ATCC BAA-999; or *L. acidophilus*, *B. bifidum*, *B. animalis* subsp. *lactis*, and *B. longum* subsp. *longum*); *B. animalis* subsp. *lactis* (including DSM 15954); *L. reuteri* (DSM 17938 or ATCC 55730); or *L. rhamnosus* (ATCC 53103 or ATC A07FA or LCR 35) for the prevention of NEC over no and other probiotics is recommended [183].

Also, reports highlighting scientific advice on the evaluation of the safety of probiotics, general guidelines for their evaluation, and specific questions related to their pathogenicity, toxicity, and allergenicity, as well as their functional and nutritional properties, were preliminarily prepared following joint consultations of the Food and Agriculture Organization (FAO) of the United Nations and the World Health Organization (WHO), i.e., FAO/WHO experts in 2001 (developed in the form of a guide by a group of experts in 2002) [184,185].

## 7. Fermented Food—Sources of Probiotics for Human Health

Around the world, people are increasingly concerned about a healthier lifestyle and are looking for functional foods. Functional foods are products that provide some benefits to humans, such as protecting the body and even preventing disease [186]. 

Fermented foods are products obtained using living microorganisms, but there is a risk that these microorganisms will not survive in the final food product due to the processing steps used in their manufacture. Food fermentation is a simple and natural method that has been used since ancient times to promote good digestion, preserve food, and improve health. It has also been used because bacteria in fermented food increase the levels of vitamins B, C, and K and neutralize harmful nutrients, such as phytic acid and protein inhibitors that release nutrients from the food that would otherwise have passed through the intestines undigested [178]. However, some homemade fermented foods, such as yogurts and pickles, are sources of probiotics and have been analyzed in clinical trials in which health benefits have been studied. Probiotics are currently available in two main forms: food and dietary supplements, and dietary supplements are regulated by the Food and Drug Administration (FDA) [178].

Most probiotics are found in milk-based products but are also found in other foods such as juices and plant-based drinks. All these products represent a challenge for people who want to eat as healthy as possible, but also for the growing vegan consumer market, which is looking for products without ingredients of animal origin. In addition, other factors including cholesterol and the presence of lactose in milk and dairy products, as well as other issues such as durability, make these a major target for the food industry, especially functional foods, both from an innovative and economic perspective.

### 7.1. Natural Sources

**Yogurt** is a dairy product obtained through the bacterial fermentation of milk. Across the world, cow’s milk is the most used milk to obtain yogurts, whereas, in various parts of the world, different types of milk are used, such as water buffalo, goat, sheep, horse, camel, and yak. *L. bulgaricus* and *S. salivarius* subsp. (thermophilic bacteria) cultures are used to make yogurts from milk, but during processing, other *lactobacilli* and *bifidobacteria* can sometimes be added. The health benefits include preventing osteoporosis and reducing the risk of high blood pressure, antibiotic-associated diarrhea and acute diarrhea in children, vaginal yeast and bacterial infections, and urinary tract infections [187].

**Cheeses** are also a source of probiotics, and sometimes, keywords such as “probiotics”, “active culture”, or “live culture” can be seen on their packaging. Some cheeses are inoculated with live cultures of *L. acidophilus* and *bifidobacterium*. Other cheeses that are good sources of probiotics are blue cheese and aged cheeses [188,189]. 

**Kefir** originates from the Caucasus and Turkey, but it is also consumed in areas such as Eastern Europe and Southwest Asia. It is a fermented milk drink that contains bacteria, such as *L. acidophilus*, *brevis*, *casei*, *delbrueckii* subsp. *Bulgaricus*, and yeasts such as *Candida humilis*, *Kazachstania unispora*, and *Kluyveromyces lactis* [190].


**Whey and buttermilk**


Whey is the translucent yellowish-green liquid fraction that remains after the coagulation of milk and removal of casein during the cheese- or casein-making process. Buttermilk is the liquid that remains after extracting butter from cream. Today, it is produced from cow’s milk using either *S. lactis* or *L. bulgaricus* [191,192].

**Koumiss**, also known by other names such as kumiss, kymmyz, coomys, kumis, kymis, and airag, is an acid-alcoholic drink widely consumed in the Middle East. This product is unique compared with other dairy products since it is produced from mare’s milk, and the cultures used in the fermentation process include various lactic bacteria and yeasts. The health benefits include a decrease in intestinal and pancreatic lipase activity, an impact on intestinal immune function in immunocompromised conditions, and a reduction in total cholesterol and LDL-cholesterol [193].

**Kombucha** is an assortment of fermented tea and is prepared from yeast, sugar, and black tea. The bacteria from the yeast proliferate in the environment with sugar content, forming a film on the surface of the drink; this film can be harvested and used to make more kombucha. During the fermentation process, acetic acid is also produced along with small amounts of alcohol and gases that give a slightly acidic taste. The yeast component of kombucha may contain *S. cerevisiae*, *Brettanomyces bruxellensis*, or *Candida stellata*, and the bacterial component consists of *Gluconacetobacter xylinu*. A few of its health benefits include antidiabetic activity, reduced atherosclerosis, antihypertensive and anti-inflammatory effects, and the alleviation of arthritis, rheumatism, and gout [194].

**Tempeh** or tempe is a traditional Indonesian food made from fermented soybeans. It is made with a controlled fermentation process that binds soybeans into a cake with the white mycelium of the spore *Rhizopus oligosporus*, forming a solid cake that can be sliced, fried, or steamed as an alternative to meat [195].

**Natto** is a traditional Japanese food based on fermented soybeans, where soybeans are steamed and then fermented with *B. subtilis* [195].

**Kvass** is a fermented beverage made from wheat, rye, barley, or rye bread. It is sometimes flavored with fruit, berries, raisins, or birch sap collected in the early spring [196].

**Bors** is the liquid obtained by the natural fermentation of an aqueous suspension of wheat bran (“tărațe” means “bran” in Romanian) or rye and corn flour [197].

**Boza** is a refreshing drink with a sweet–sour taste, which is obtained by boiling ground millet with water. It is a drink of the nomadic Turks from Central Asia, recorded since the 10th century, which then reached Anatolia, and from there it spread, including into the Balkans. The fermentation of the drink begins with a weak alcoholic fermentation which, in a short time, is replaced by acid fermentation of the lactic, acetic, butyric, etc., types. Braga is a complex environment, conducive to the development of various microorganisms, most of which are represented by *E. coli* and *Saccharomyces*, which give off a pronounced particular aroma. There are also cocci, bacilli, and molds of the *Mucor type* [198].

**Miso** is a traditional Japanese food. It is a thick paste made by fermenting soybeans with salt and koji and sometimes rice, barley, seaweed, or other ingredients. The bacteria found in miso include *Tetragenococcus halophilus* and *L. acidophilus* [195].

**Sauerkraut** is finely chopped raw cabbage that has been fermented by various lactic acid bacteria. It has a long shelf life and a distinct sour flavor, both of which result from the lactic acid formed when the bacteria ferment the sugars in the cabbage leaves. The word “sauerkraut” comes from German, meaning “sour plants”, and the Korean version is known as “kimchee”. Plant fermentations contain four species of lactic acid bacteria: *Leuconostoc mesenteroides*, *L. brevis*, *L. plantarum*, and *Pediococcus pentosaceus*. Kimchi also contains *L. kimchii*. The health benefits include increasing the bioavailability of nutrients, promoting gut health, cardioprotective action, stimulating the immune system, antihyperlipidemic activity, anti-inflammatory activity, and anticancer properties [199].

**Soy sauce** is a traditional Asian condiment obtained by fermenting a soybean paste with salt and enzymes. This special spice appeared approximately 2200 years ago in the west of Ancient China during the Han Dynasty, where it was used to season various dishes. In the fermentation process of soybeans, *Aspergillus oryzae* or *Aspergillus sojae* molds are used together with water and salt. Soy sauce is also rich in lactic acid bacteria and has multifarious health benefits, such as lowering serum cholesterol and anti-diabetic, anti-hypertensive, anti-cardiovascular, and anti-neuroinflammatory effects [195].

Eating fermented foods has traditionally been associated with health benefits. In epidemiological studies, fermented dairy products have been associated with a reduced risk of metabolic syndrome [186], a reduced risk of obesity [200], a reduced risk of cardiovascular disease [201], and a reduced risk of colon cancer [202,203]. The consumption of fermented soybeans (miso and natto) has been associated with a reduced risk of hypertension [204] and a reduced cardiovascular risk [205].

**Pickles** (also known as gherkins) are cucumbers that have been preserved in a solution of salt and water. They are left to ferment for some time, using their own naturally present lactic acid bacteria. This process makes them sour. Pickled cucumbers are a great source of healthy probiotic bacteria, which may improve digestive health. They are also low in calories and a good source of vitamin K, an essential nutrient for blood clotting [206].

**Sour red soup** is a traditional fermented product from southwest China. It is produced by the natural fermentation of a mixture consisting of tomatoes, red cayenne, glutinous rice flour, white wine, and other ingredients, being rich in a variety of organic acids, minerals, and other nutrients. It has an important role in maintaining nerve and muscle excitability and in maintaining the body’s acid–base balance [207].

**Brem** is a traditional food made from white glutinous rice that is originally from Indonesia [208]. Brem is produced from glutinous rice extract, has a sweet–sour taste and starchy texture, and is usually eaten as a snack. It is good for skin health, warming the body, and increasing appetite [209].

### 7.2. Other Sources

All over the world, human populations are looking for a healthier diet. This has led to the development of new foods that have functional properties: they contain probiotics, prebiotics, or symbiotics. Other sources of probiotics are different categories of products, such as medicines, cosmetics, and food supplements.

Probiotics used in cosmetic products aim to bind them to the epidermal surface, inhibit pathogens, produce antimicrobial substances, and also increase the immunomodulatory properties. Probiotic strains that are contained in cosmetic products include *B. subtilis*, *L. acidophilus*, *L. casei*, *L. lactis*, and *L. plantarum*. Regarding the usage of probiotics in cosmetology and dermatology, they are mainly incorporated into products for washing and caring for skin with atopic dermatitis; for treating acne-prone skin and skin with eczema and psoriasis; and even for skin after invasive treatments in cosmetology or medicine (exfoliation with acids) [210].

In cosmetology, probiotics have been added to everyday care products. Fragments of cell walls and non-living bacteria are contained in probiotic cosmetics such as serums, creams, ointments, body gels, body balms, and shampoos [210].

Food (beverages, drinks, and desserts) obtained with fermentation is the main food matrix because fermentation ensures greater viability for the probiotic culture. Beverages produced with cereals, pseudocereals, and seeds (for example, barley, oats, millet, corn, sorghum, rice, quinoa, and chia) have been formulated to obtain new probiotic products. Also, tea and fruit juices are interesting new matrices that have been studied for the addition of probiotics because they are composed of bioactive compounds, such as vitamins, minerals, and polyphenols. Morais et al. [211] stated that in the case of using fruits as a food matrix, the metabolism of probiotics can increase the bioaccessibility of phenolic compounds and can increase functionality. Worldwide, there is a multitude of food products based on probiotics and supplements/medicines, some examples include Avenly vele (an oat-based drink), produced by Avenly Oy Ltd.; Kind Breakfast Bars (flavored cereal breakfast bars), produced by Kind; Probiotic Muffin (muffin), produced by Isabella’s Healthy Bakery; Probio Yogurt (coconut milk and red fruits), produced by Puravida; Good Cacau (Chocolat), produced by Be Good; VSL#3 (capsules), produced by Alfasigma; Protectis (chewable tablets or probiotic drops), produced by Biogaia; Tipton Mills Probiotic Coffee (instant coffee), produced by Tipton Mills; and Sho Balance (vegan spheric-gels), produced by Sho Nutrition [212]. Probiotics can consist of one or a combination of a few strains, either as capsules, powders, or as a component of a food. The ways in which the existing strains could contribute to human health are provided in Table 2.

According to [229], different probiotics have been administered as supplements under controlled conditions to many patients (adults and children) suffering from various diseases and have been shown to be beneficial without any associated risk.

## 8. Recent Advances and New Trends

Probiotics are crucial in maintaining the balance of the human intestinal microbiota, and until now numerous scientific reports have confirmed their positive effect on human health. Probiotic microorganisms are attributed a high therapeutic potential in acute diarrheal disease, irritable bowel syndrome, bacterial and viral infections (HIV, HPV), obesity, insulin resistance syndrome, type 2 diabetes and NAFLD, cancer, allergies, neurological diseases, lactose intolerance, and immunity, and they are crucial for the maintenance of the balance of human intestinal microbiota. Until now, probiotics have been used to treat various medical problems, but recent advances in culturomics, such as the use of gnotobiotic animal models, have led to a new field in the development of new host-specific probiotic drugs. A new trend in the production of probiotics is associated with the design of probiotics. Designer probiotics are commensal strains of bacteria that are engineered or modified so that they can withstand the myriad stresses they encounter both outside and inside our bodies (freeze-drying/manufacturing/acids/temperature) or simply to improve functions most beneficial to the host. Synthetic biology has enabled the engineering of both probiotic and commensal strains to acquire and perform new functions.

Another trend in the medical world is fecal microbiota transplantation (FMT), also known as fecal/stool transplantation or fecal bacteriotherapy. The basic principle of fecal transplantation consists of restoring the microbiome balance in the patient’s intestine by transferring healthy bacteria (probiotic strains). It helps to restore the balance between beneficial and harmful bacteria, thus restoring intestinal health. This type of treatment is for a specific condition involving the infusion or grafting of filtered excrement from any healthy donor into the recipient’s stomach. Studies have already been described in the literature in which FMT plays a role in the treatment of inflammatory and infectious skin diseases such as psoriatic arthritis, psoriasis, or alopecia universalis [230].

Considering that there is a great variation regarding the intestinal microbiota in humans, it is possible that soon, a person’s optimal diet will be adapted to the intestinal microbiota, and nutrigenomics and nutrigenetics will play an important role in establishing a person’s diet.

A new emerging trend is to use next-generation probiotics. Next-generation probiotics are outside of the commonly used probiotic spectrum (*Lactobacilli*, *Bifidobacteria*, etc.), and large-scale genomic testing has identified probiotic strains with potential health benefits from the genera *Bacteroides*, *Akkermasia*, *Faecalibacterium*, and *Eubacterium* [231]. These new candidates represent a significant proportion of the currently cultivable human gut microbiome. Also, these genera can provide physiological functions that are not always conferred directly by *bifidobacteria* or *lactobacilli* (the production of butyrate, propionate, and other bioactive) [232]. Transforming these species into industrially viable probiotics presents challenges, as new costs are added due to their requirements of rich growth media and anaerobic conditions, as well as the investment in determining optimal fermentation and manufacturing processes. *A. muciniphila* is one of the most promising candidates despite all these difficulties [3].

Regarding probiotic foods, future sources of fermented/unfermented food matrices may include not only fruits, vegetables, grains/cereals, and dairy products but also meat and fish products and honey, as well as environmental sources such as soil [233]. It is very important that the use of probiotics in food/nutrition presents evidence regarding their safety and efficacy, and the necessary proof must come from scientific studies demonstrating a history of safe use, the absence of virulence and pathogenic factors, the absence of the production of substances or metabolites that cause health risks to humans, and the absence of adverse events. However, new advances in biotechnology and bioinformatics will provide new detailed insights into the action of probiotics, as well as clues for the identification of new candidate organisms and substrates. These new discoveries and validation techniques will continue to be refined, increasing the reliability and reproducibility of results from in vitro and in vivo studies. This will allow better comparability of datasets and new perspectives focused on several research directions. All these insights, as well as population-based studies, will lead to the discovery of new ways to improve dietary relevance and clinical efficacy, as well as adapt to everyone’s specific biology and microbiome. Such a vision is the predicted future of probiotics.

## 9. The Side Effects and Risks of Probiotics

Live microorganisms in fermented foods have been used for centuries without causing human disease. Probiotic products administered as dietary supplements are regulated as foods, not as drugs [178]. Probiotic supplements (containing a variety of microorganisms in powder, pill, or liquid form) are safe for most people to take daily. Although probiotics have a multitude of benefits, some people have experienced mild side effects, many of which will resolve as the body’s microbiome readjusts [234]. Common probiotic side effects are bloating, gas, diarrhea, constipation, nausea [235,236], thirst, headaches and migraines, and skin reactions [234]. Mild gastrointestinal symptoms (gas and bloating), usually improve within a few weeks [236]. Some people experience increased thirst when they start taking yeast probiotics, especially in the first week as their bodies adjust to the new balance of gut bacteria. Symptoms go away on their own for most people [234]. Some probiotic foods, especially fermented foods (e.g., sauerkraut and kimchi), contain high levels of biogenic amines, including histamine, tyramine, tryptamine, and phenylethylamine, which are compounds that can trigger headaches and migraines in some people. Also, biogenic amines were found in some yogurts, but to a much lower degree [234]. Very rarely, probiotics can cause rashes or itchy skin. This may be due to an allergic reaction to an ingredient in the supplement. It will usually disappear shortly after the person stops taking the supplement [234].

In addition to side effects, there is also the risk that probiotics (being living organisms) may cause infections that must be treated with antibiotics, especially in people with underlying conditions. To obtain their health benefits, many of the probiotic strains are genetically modified in the laboratory. For this reason, the safety of each strain must be guaranteed and strictly monitored so that they cannot accumulate in the environment, possess selection markers for antibiotics, or transfer any harmful genetic information to other bacteria [178]. In conclusion, probiotic supplementation should be used with caution in persons at risk. These include:People on anti-rejection medication after a stem cell or solid organ transplant.People on immunosuppressive medication or corticosteroids and chemotherapy for cancer, and people with autoimmune diseases.People with structural heart disease with valvular abnormality or valve replacement or a history of infective endocarditis.People with acute abdomen, active intestinal disease including colitis and the presence of neutropenia or the anticipation of neutropenia after chemotherapy and radiotherapy or the presence of active intestinal perforation and leakage [237].

## 10. Conclusions

The evidence for the role of the microbiome/microbiota in human health (acute and chronic) is now relevant, with the specification that the influence of our microorganisms goes beyond the gut to incorporate the brain, metabolism, and immune system. While aging and genetics influence the composition and diversity of the microbiome, other modifiable factors, such as diet, exercise, exposure to exogenous microbes, and antibiotic use, may be more important for achieving microbiota eubiosis. While this offers people the chance to adopt personalized lifestyles, there are also several challenges, including gathering adequate evidence to establish which microbiota interventions are appropriate for which population groups, understanding the mechanisms involved in these processes, and developing new effective probiotic products. Reviewing the literature, there is a tantalizing opportunity to find ways to live in harmony with our microbiome through the consumption of probiotics or fermented foods/supplements that have probiotics included in the matrix, which may provide large-scale health benefits. Probiotics are the elements that provide an effective strategy to prevent or ameliorate many diseases. However, future research should obtain evidence regarding the optimal dose for each in-dividual strain and the most effective matrix. Future studies should also determine whether probiotic strains, be them ancient or of the new generation, colonize our intestines, or they are simply transitory microorganisms with beneficial effects.

## Figures and Tables

**Figure 1 microorganisms-12-00234-f001:**
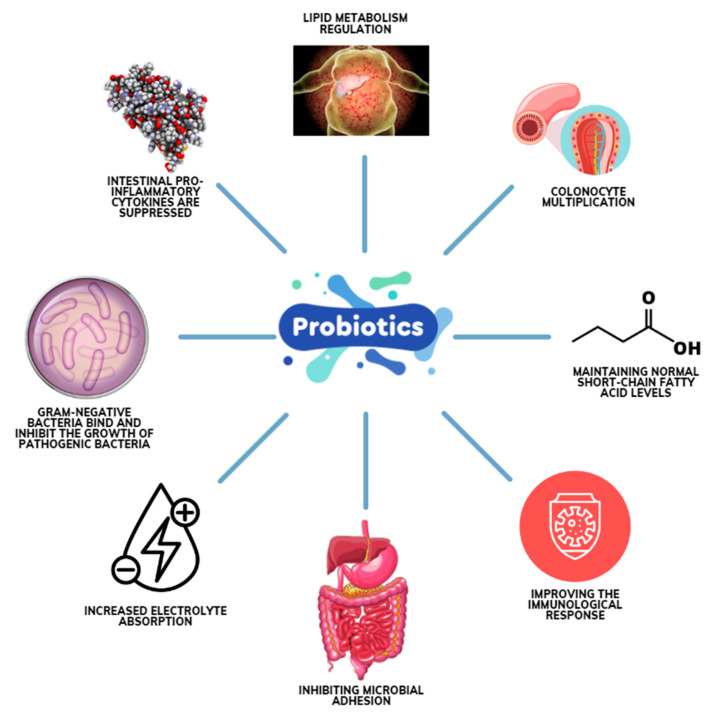
Action mechanisms of probiotics.

**Table 1 microorganisms-12-00234-t001:** Functions of probiotics.

Type	Strain	Function	References
** *Lactobacillus* **	*Lactobacillus acidophilus*	Used in the treatment of diarrhea and for relief from the symptoms of irritable bowel syndrome; can reduce cytokines to relieve inflammatory bowel disease, modulate immunity, lower cholesterol, relieve diarrhea, and alleviate cancer.	[36,37,38]
	*Lacticaseibacillus casei*	Prevention or treatment of diseases that disrupt the intestinal microbiota: diarrhea and constipation, relief from the symptom of irritable bowel syndrome, and gingivitis, and the anti-inflammatory response.	[39,40,41,42]
	*Lactobacillus rhamnosus*	Has anti-diabetic and anti-viral activity; used to treat obesity; has the ability to fight against pathogenic bacteria and fungi in the genitourinary tract, preventing the recurrence of urinary tract infections in postmenopausal women; prevents enteric colonization by *Candida* species and treats recurrent *Clostridium difficile*-induced colitis in children; and has an effect on symptoms of maternal depression and anxiety during the postpartum period.	[43,44,45,46,47,48]
	*Lactiplantibacillus plantarum*	Has the ability to prevent the production of endotoxin; improves of the symptom of irritable bowel; and has antimicrobial activity and cholesterol lowering activity.	[49,50]
	*Lactobacillus reuteri*	Treatment of gingivitis in pregnant women and chronic periodontitis.	[50,51,52,53,54]
** *Bifidobacterium* **	*Bifidobacterium infantis*	Can improve the symptoms of irritable bowel syndrome and inhibits the secretion of allergen induced IgE.	[55,56,57,58]
	*Bifidobacterium adolescentis*	Can reduce inflammation of the spleen and brain and changes the microbiota of cecum and colon.	[59]
	*Bifidobacterium bifidum*	Can reduce cholesterol; is used in the treatment of infant diarrhea; and has greater cytokine (IL-6) production and active phagocytic properties.	[60,61,62]
	*Bifidobacterium longum*	Used in treating diarrhea and provides relief from the symptoms of irritable bowel syndrome; modulates the immune system through IL-10 production.	[63,64]
**Other Lactic Acid Bacteria**	*Streptococcus thermophilus*	Can produce antioxidant compounds and mitigate the risk of some types of cancer; has anti-inflammatory, antimutagenic effects and stimulates the gut immune system; and is useful in inflammatory bowel disease.	[65,66]
	*Enterococcus faecium*	Modulates the Th2-mediated pathologic response.	[67,68]
**Other Microorganisms**	*Bacillus subtilis*	Used in treating diarrhea, in the eradication of *H. pylori*, and in the production of vitamin K.	[69,70]
	*Bacillus coagulans*	Can regulate the balance of intestinal microbiota and improve immunity; promotes the metabolism and utilization of nutrients; and is able to resist high temperatures and has acid and bile resistance.	[71,72,73]
	*Saccharomyces boulardi*	Used in treating diarrhea, ulcerative colitis, and a symptom of irritable bowel syndrome.	[74,75,76]
	*Escherichia coli NISSLE 1917*	Used in the treatment of intestinal diseases (diarrhea, inflammatory bowel disease, ulcerative colitis); can exhibit antagonistic effects on a variety of intestinal pathogenic bacteria; regulates the secretion of immune factors in vivo; and enhances the ability of host immunity.	[77,78,79]

**Table 2 microorganisms-12-00234-t002:** Probiotics from food and their benefits.

Genus	Found in the Body	Dietary Source	Potential Benefits	References
*Lactobacillus*	Colon, gut, and vagina	Yogurt, fermented foods, bread, sauerkraut, wine, cereals (oat bran, whole grain etc.)	Gastroenteritis, easing lactose intolerance, immune system modulation, cancer protection, modulating brain activity, alleviating inflammation, lowering cholesterol, preventing pathogen colonization, bile resistant	[213,214,215,216,217,218]
*Bifidobacterium*	Breast milk, oral cavity, colon, and vagina	Yogurt, kefir, kombucha, sauerkraut	Bile resistant, antibiotic-associated diarrhea, eczema, immune system modulation, cholesterol-lowering abilities	[219,220,221,222]
*Saccharomyces*	Decaying fruit, plants, soil, insects, colon	Yogurt, wine, kombucha, sauerkraut	Antibiotic-associated diarrhea, preventing recurring *Clostridium difficile* infections, irritable bowel syndrome, travelers’ diarrhea	[223,224,225]
*Escherichia coli*	Colon	Capsules	Production of defensin, tight-junction protein modification, irritable bowel disorder, constipation, pro-inflammatory properties, antagonistic properties against a variety of pathogens, colon cancer	[226,227,228]

## Data Availability

Not applicable.

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
