# Peer review of "The Potential Impact of Probiotics on Human Health: An Update on Their Health-Promoting Properties"

_microorganisms, 2024, doi:10.3390/microorganisms12020234_

Round 1

Reviewer 1 Report

Comments and Suggestions for Authors

The paper is interesting but quite superficial. Major revisions need to be made before publishing:

1. In "Table 1. Functions of probiotics", the authors quote 1-2 references for a specific species and present very basic functions. There are no more specialized publications and therefore no functions, e.g. anti-caries, effects in periodontal diseases, anticancer. The table should include information whether a given species is used in humans or only in veterinary medicine.

2. In the chapter "4. Mechanisms of action of probiotics", there is actually no information about the mechanisms. Mechanisms of action include, for example, action through lactic acid, hydrogen peroxide, or bacteriocins and influence on pH, production of cytokines, apoptosis, antagonism towards anaerobic bacteria, etc. Table 2 and the descriptions in this chapter do not match the title.

3. In the chapter "5. Probiotics and human health", please refer to the guidelines on wound treatment, gastroenterology, etc.

4. Kefirs do not only come from the Caucasus and Turkey. They are also common in Slavic countries, such as Poland, Belarus, and Ukraine.

5. References are missing in chapter "6.1. Natural sources". Where did the authors get this data? The distribution also lacks many probiotic products, such as pickled cucumbers, juices, and sour soup.

6. I propose to add a section on the side effects of probiotics. From clinical practice, I see that they are common, e.g. in patients with inflammatory bowel disease using Saccharomyces boulardii. Therefore, in many European countries, the use of S. boulardii in patients with intestinal diseases is being abandoned.

Comments on the Quality of English Language

Small corrections required.

Author Response

Thank you for the time given to the review and we respond promptly to the comments, because we consider that all observations are objective and well founded.

We also mention that we tried to improve our manuscript and we hope that we succeeded, but we took into account the suggestions of all the reviewers.

Point 1: In "Table 1. Functions of probiotics", the authors quote 1-2 references for a specific species and present very basic functions. There are no more specialized publications and therefore no functions, e.g. anticaries, effects in periodontal diseases, anticancer. The table should include information whether a given species is used in humans or only in veterinary medicine

Response 1: We made the necessary changes

Point 2: In the chapter "4. Mechanisms of action of probiotics", there is actually no information about the mechanisms. Mechanisms of action include, for example, action through lactic acid, hydrogen peroxide, or bacteriocins and influence on pH, production of cytokines, apoptosis, antagonism towards anaerobic bacteria, etc. Table 2 and the descriptions in this chapter do not match the title.

Response 2: I redid according to the requirements

Point 3: In the chapter "5. Probiotics and human health", please refer to the guidelines on wound treatment, gastroenterology, etc.

Response 3 The changes have been made.

Point 4: Kefirs do not only come from the Caucasus and Turkey. They are also common in Slavic countries, such as Poland, Belarus, and Ukraine.

Response 4: Correct. I made the necessary changes.

Point 5: References are missing in chapter "6.1. Natural sources". Where did the authors get this data? The distribution also lacks many probiotic products, such as pickled cucumbers, juices, and sour soup.no correlation between the way the results are presented in the Fig. 1 and their description in the text,

Response 5: I have added references from literature.

Point 6: I propose to add a section on the side effects of probiotics. From clinical practice, I see that they are common, e.g. in patients with inflammatory bowel disease using Saccharomyces boulardii. Therefore, in many European countries, the use of S. boulardii in patients with intestinal diseases is being abandoned

Response 6: I added information about the side effects of probiotics 

For all suggestions for corrections which were introduced in the attached pdf file, I have modified the text where it was suggested. All changes have been marked in red.

Reviewer 2 Report

Comments and Suggestions for Authors

The document titled: “The potential impact of probiotics in human health: An update on their health promoting properties” shows a very general review of the probiotics. In addition, there is not depth of the topic on the mechanisms that these microorganisms use to provide benefits. On the other hand, they mention a description of the foods that contain these microorganisms, when this is widely reported. Authors should review the English language.

The introduction is very general, paragraph four (lines 73-95) does not mention the properties that probiotics have to provide benefits health. Furthermore, not genera or scientific names are mentioned, that allow the identification of microorganisms with this potential.

It is important to mention that probiotics are not only used in fermented foods, they are also used to make supplements and other products, so it is important to review line 80-82.

Is it necessary to include a materials and methods section in a review?

Line 117-132: information not relevant for the topic of the document.  

Line 135-136: Groups should not be written in italics, genera or scientific names should.

Line 137-141: Information on the Lactobacillus genus is very scarce compared to that is mentioned for Bifidobacterium.

In Table 1 Bifidobacteria is mentioned as a strain, it gives some characteristics, which could be written when characteristics of the group are mentioned.

The information in Table 2 is not relevant, it can be commented on in the paragraph.

Figure 1 could be represented through images, since it is very simple to display an image with text boxes.

In section 6.1 Natural sources (Line 558-628), unnecessary information is described regarding foods that contain probiotics, and no references are given.

Line 671-686, 696-719, there are no references that support the information.

Check the author guide for how to cite references.

Comments on the Quality of English Language

 Authors should review the English language.

Author Response

Point 1: The introduction is very general, paragraph four (lines 73-95) does not mention the properties that probiotics have to provide benefits health. Furthermore, not genera or scientific names are mentioned, that allow the identification of microorganisms with this potential.

It is important to mention that probiotics are not only used in fermented foods, they are also used to make supplements and other products, so it is important to review line 80-82.

Response 1: We made the necessary changes

Point 2: Is it necessary to include a materials and methods section in a review?

Response 2: They are not necessary, so I removed them from the manuscript

Point 3: Line 117-132: information not relevant for the topic of the document. 

Response 3 The changes have been made.

Point 4: Line 135-136: Groups should not be written in italics, genera or scientific names should.

Response 4: Correct. I made the necessary changes.

Point 5: Line 137-141: Information on the Lactobacillus genus is very scarce compared to that is mentioned for Bifidobacterium.

Response 5: Correct. I made the necessary changes.

Point 6: In Table 1 Bifidobacteria is mentioned as a strain, it gives some characteristics, which could be written when characteristics of the group are mentioned.

Response 6: I added the necessary information

Point 7: The information in Table 2 is not relevant, it can be commented on in the paragraph.

Response 7: Correct. I've done the modification

Point 8: Figure 1 could be represented through images, since it is very simple to display an image with text boxes.

Response 8: I've done the modification

Point 9: In section 6.1 Natural sources (Line 558-628), unnecessary information is described regarding foods that contain probiotics, and no references are given.

Response 9: I added references

Point 10: Line 671-686, 696-719, there are no references that support the information.

Response 10: I added references

For all suggestions for corrections which were introduced in the attached pdf file, I have modified the text where it was suggested. All changes have been marked in red.

Reviewer 3 Report

Comments and Suggestions for Authors

In this manuscript, the authors summarized the fermented food matrix around probiotic bacteria plays an important role in the survival of these strains and maintaining health of the microbiome. Among them, fermented foods have been associated with the prevention of irritable bowel syndrome, lactose intolerance and gastroenteritis, obesity, but also in other conditions such as chronic diarrhea, allergies, dermatitis, bacterial and viral infections, all of which are closely related to an unhealthy lifestyle. Recent and ongoing developments in microbiome/microbiota science allow us new research directions for probiotics. The new types, mechanisms and applications studied so far and those currently under study have great potential to change scientific understanding and nutritional applications and human health care. The expansion of related fields related to the study of the microbiome and the involvement of probiotics in its improvement foreshadows an era of significant changes.

This article tries to analyze recent, emerging, and anticipated trends in probiotics and create a vision for related areas of influence development in the field. However, there are still some issues that need to be revised in this article:

1.Line 36-39. “Torres et al. [1] reported that the role of a balanced diet (with the aim of maintenance of human health) is the principal interest of the scientific community, and numerous research have proven the reduce inthe risk of the appearance of some diseases by the consuming of some probiotics-based foods.”.

Please reorganize the expression in the introduction. Please refer this reference(Food & Function, 2022, 13(24), 12686-12696).

2. Line68-72. It should be reorganization about diet and oxidative stress. Please refer this reference (Critical Reviews in Food Science and Nutrition, 2023, 63(29), 9816–9842.).

3. 6.1. Natural sources

There still have fermentation source. Please refer this reference (Food chemistry. 437(2024), 137834.).

4. The figure about fermented food matrix around probiotic bacteria with human body should be summarized.

5. The expression should be fluence.

6. The reference should be updated in recent years.

Comments on the Quality of English Language

In this manuscript, the authors summarized the fermented food matrix around probiotic bacteria plays an important role in the survival of these strains and maintaining health of the microbiome. Among them, fermented foods have been associated with the prevention of irritable bowel syndrome, lactose intolerance and gastroenteritis, obesity, but also in other conditions such as chronic diarrhea, allergies, dermatitis, bacterial and viral infections, all of which are closely related to an unhealthy lifestyle. Recent and ongoing developments in microbiome/microbiota science allow us new research directions for probiotics. The new types, mechanisms and applications studied so far and those currently under study have great potential to change scientific understanding and nutritional applications and human health care. The expansion of related fields related to the study of the microbiome and the involvement of probiotics in its improvement foreshadows an era of significant changes.

This article tries to analyze recent, emerging, and anticipated trends in probiotics and create a vision for related areas of influence development in the field. However, there are still some issues that need to be revised in this article:

1.Line 36-39. “Torres et al. [1] reported that the role of a balanced diet (with the aim of maintenance of human health) is the principal interest of the scientific community, and numerous research have proven the reduce inthe risk of the appearance of some diseases by the consuming of some probiotics-based foods.”.

Please reorganize the expression in the introduction. Please refer this reference(Food & Function, 2022, 13(24), 12686-12696).

2. Line68-72. It should be reorganization about diet and oxidative stress. Please refer this reference (Critical Reviews in Food Science and Nutrition, 2023, 63(29), 9816–9842.).

3. 6.1. Natural sources

There still have fermentation source. Please refer this reference (Food chemistry. 437(2024), 137834.).

4. The figure about fermented food matrix around probiotic bacteria with human body should be summarized.

5. The expression should be fluence.

6. The reference should be updated in recent years.

Author Response

Thank you for the time given to the review and we respond promptly to the comments, because we consider that all observations are objective and well founded.

We also mention that we tried to improve our manuscript and we hope that we succeeded, but we took into account the suggestions of all the reviewers.

Point 1: Line 36-39. “Torres et al. [1] reported that the role of a balanced diet (with the aim of maintenance of human health) is the principal interest of the scientific community, and numerous research have proven the reduce in the risk of the appearance of some diseases by the consuming of some probiotics-based foods.”.

Please reorganize the expression in the introduction. Please refer this reference(Food & Function, 2022, 13(24), 12686-12696).

Response 1: I modified and added the indicated reference to the manuscript.

Point 2: Line68-72. It should be reorganization about diet and oxidative stress. Please refer this reference (Critical Reviews in Food Science and Nutrition, 2023, 63(29), 9816–9842.).

Response 2: I modified and added the indicated reference to the manuscript.

Point 3: 6.1. Natural sources. There still have fermentation source. Please refer this reference (Food chemistry. 437(2024), 137834.).

Response 3: The changes have been made.

Point 4: The figure about fermented food matrix around probiotic bacteria with human body should be summarized.

Response 4: I made the necessary changes.

Point 5: The expression should be fluence.

Response 5: I revised the English language.

Point 6: The reference should be updated in recent years.

Response 6: I have updated the references, the number of references from the last 5 years and older than five years corresponds to the journal requirements

For all suggestions for corrections which were introduced in the attached pdf file, I have modified the text where it was suggested. All changes have been marked in red.

Reviewer 4 Report

Comments and Suggestions for Authors

This review summarized recent discoveries about the health promoting properties of probiotics. This is a meaningful topic and there is abundance of information in the manuscript. However, I feel that the topic is too big and some discussions are superficial. The authors should better narrow down the topic and discuss in depth some certain aspects. The format of the manicurist also needs improvement.

Author Response

Thank you for the time given to the review and we respond promptly to the comments, because we consider that all observations are objective and well founded.

We also mention that we tried to improve our manuscript and we hope that we succeeded, but we took into account the suggestions of all the reviewers.

Point 1: This review summarized recent discoveries about the health promoting properties of probiotics. This is a meaningful topic and there is abundance of information in the manuscript. However, I feel that the topic is too big and some discussions are superficial. The authors should better narrow down the topic and dis Biocatalysis and Agricultural Biotechnology 53 (2023) 102889

Response 1 I consulted the indicated article and also reorganized and improved the information in the manuscript.

Round 2

Reviewer 2 Report

Comments and Suggestions for Authors

The comments were done

Author Response

Thank you!

Reviewer 3 Report

Comments and Suggestions for Authors

It can be accepted in the current revision.

Comments on the Quality of English Language

It can be accepted in the current revision.

Author Response

Thank you!